# DO EMBODIED AGENTS DREAM OF PIXELATED SHEEP?:

# EMBODIED DECISION MAKING USING LANGUAGE GUIDED WORLD MODELLING

**Kolby Nottingham** [* †]   **Prithviraj Ammanabrolu** [‡]   **Alane Suhr**[‡]

**Yejin Choi**[‡ §]   **Hannaneh Hajishirzi**[‡ §]   **Sameer Singh**[† ‡]   **Roy Fox**[†]

## ABSTRACT

Reinforcement learning (RL) agents typically learn tabula rasa, without prior knowledge of the world, which makes learning complex tasks with sparse rewards difficult. If initialized with knowledge of high-level subgoals and transitions between subgoals, RL agents could utilize this *Abstract World Model* (AWM) for planning and exploration. We propose using few-shot large language models (LLMs) to hypothesize an AWM, that is tested and verified during exploration, to improve sample efficiency in embodied RL agents. Our DECKARD agent[1] applies LLM-guided exploration to item crafting in Minecraft in two phases: (1) the *Dream* phase where the agent uses an LLM to decompose a task into a sequence of subgoals, the hypothesized AWM; and (2) the *Wake* phase where the agent learns a modular policy for each subgoal and verifies or corrects the hypothesized AWM on the basis of its experiences. Our method of hypothesizing an AWM with LLMs and then verifying the AWM based on agent experience not only increases sample efficiency over contemporary methods by an order of magnitude but is also robust to and corrects errors in the LLM—successfully blending noisy internet-scale information from LLMs with knowledge grounded in environment dynamics.

## 1  INTRODUCTION

Despite evidence that practical sequential decision making systems require efficient exploitation of prior knowledge regarding a task, the current prevailing paradigm in reinforcement learning (RL) is to train *tabula rasa*, without any pretraining or external knowledge (Agarwal et al., 2022). In an effort to shift away from this paradigm, we focus on the task of creating embodied RL agents that can effectively exploit large-scale external knowledge sources presented in the form of pretrained large language models (LLMs).

LLMs contain potentially useful knowledge for completing tasks and compiling knowledge sources Petroni et al. (2019). Previous work attempted to apply knowledge from LLMs to decision-making by generating action plans for embodied environments Ichter et al. (2022); Huang et al. (2022b); Song et al. (2022b); Singh et al. (2022); Liang et al. (2022b); Huang et al. (2022a). However, LLMs still often fail when generating plans due to a lack of grounding Valmeekam et al. (2022). Additionally, many of these agents that rely on LLM knowledge at execution time are limited in performance by the accuracy of LLM output. We hypothesize that if LLMs are instead applied to improving exploration during training, resulting policies will not be constrained by the accuracy of an LLM.

Exploration in environments with sparse rewards becomes increasingly difficult as the size of the explorable state space increases. For example, the popular 3D embodied environment Minecraft has a large technology tree of craftable items with complex dependencies and a high branching factor.

---

[*]Correspondence to knotting@uci.edu

[†]Department of Computer Science, University of California Irvine, Irvine, CA, United States

[‡]Allen Institute for Artificial Intelligence, Seattle, WA, United States

[§]Paul G. Allen School of Computer Science, Seattle, WA, United States

[1]https://deckardagent.github.io/

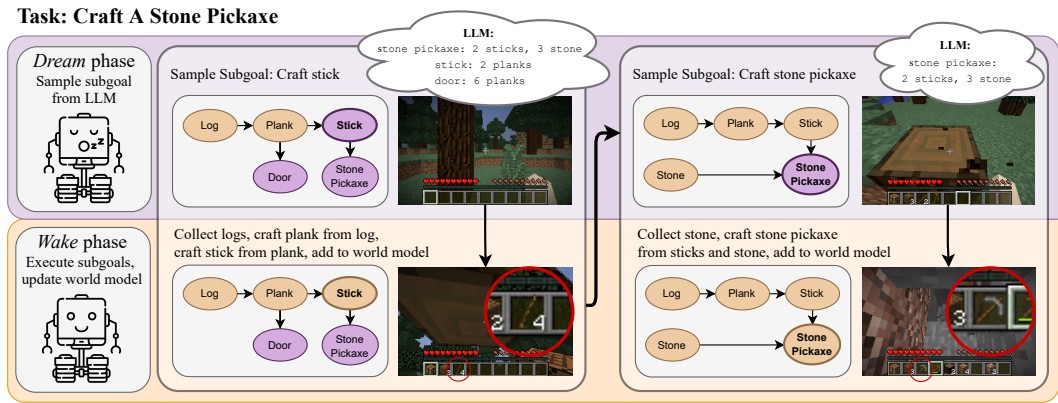

Figure 1: During the *Dream* phase, DECKARD uses the LLM-predicted DAG of subgoals, the **hypothesized** Abstract World Model (AWM), to sample a node on the path to the current task. Then, during the *Wake* phase, the agent executes subgoals and explores until reaching the sampled node. The AWM is corrected and discovered nodes marked as **verified**.

Before crafting a stone pickaxe in Minecraft an agent must: collect logs, craft logs into planks and then sticks, craft a crafting table from planks, use the crafting table to craft a wooden pickaxe from sticks and planks, use the wooden pickaxe to collect cobblestone, and finally use the crafting table to craft a stone pickaxe from sticks and cobblestone. Reaching a goal item is difficult without expert knowledge of the Minecraft crafting tree from dense rewards Baker et al. (2022); Hafner et al. (2023) or expert demonstrations Skrynnik et al. (2021); Patil et al. (2020), making item crafting in Minecraft a long-standing AI challenge Guss et al. (2019); Fan et al. (2022).

We propose DECKARD (**DEC**ision-making for **K**nowledgable **A**utonomous **R**einforcement-learning **D**reamers), an agent that hypothesizes an *Abstract World Model* (AWM) over subgoals by few-shot prompting an LLM, then exploits the AWM for exploration and verifies the AWM with grounded experience. As seen in Figure 1, DECKARD operates in two phases: (1) the *Dream* phase where it uses the hypothesized AWM to suggest the next node to explore from the directed acyclic graph (DAG) of subgoals; and (2) the *Wake* phase where it learns a modular policy of subgoals, each trained on RL objectives, and verifies the hypothesized AWM with grounded environment dynamics. Figure 1 shows two iterations of the DECKARD agent learning the "craft a stone pickaxe" task in Minecraft. During the first *Dream* phase, the agent has already verified the nodes *log* and *plank*, and DECKARD suggests exploring towards the *stick* subgoal, ignoring nodes such as *door* that are not predicted to complete the task. Then, during the following *Wake* phase, DECKARD executes each subgoal in the branch ending in the *stick* node and then explores until it successfully crafts a stick. If successful, the agent marks the newly discovered node as verified and proceeds to the next iteration.

We evaluate DECKARD on learning to craft items in the Minecraft technology tree. We show that LLM-guidance is essential to exploration in DECKARD, with a version of our agent without LLM-guidance taking over twice as long to craft most items during open-ended exploration. Whereas, when exploring towards a specific task, DECKARD improves sample-efficiency by an order of magnitude versus comparable agents, (12x the ablated DECKARD without LLM-guidance). Our method is also robust to task decomposition errors in the LLM, consistently outperforming baselines as we introduce errors in the LLM output. DECKARD demonstrates the potential for robustly applying LLMs to RL, thus enabling RL agents to effectively use large-scale, noisy prior knowledge sources for exploration.

## 2 RELATED WORK

### 2.1 LANGUAGE-ASSISTED DECISION MAKING

Textual knowledge can be used to improve generalization in reinforcement learning through environment descriptions Branavan et al. (2011); Zhong et al. (2020); Hanjie et al. (2021) or language

instructions Chevalier-Boisvert et al. (2019); Anderson et al. (2018); Ku et al. (2020); Shridhar et al. (2020). However, task specific textual knowledge is expensive to obtain, prompting the use of web queries Nottingham et al. (2022) or using models pretrained on general world knowledge Dambekodi et al. (2020); Suglia et al. (2021); Ichter et al. (2022); Huang et al. (2022b); Song et al. (2022b).

LLMs can also be used as an external knowledge source by prompting or finetuning them to generate action plans. However, by default, the generated plans are not grounded in environment dynamics and constraining output can harm model performance, both of which lead to subpar performance of out-of-the-box LLMs on decision-making tasks Valmeekam et al. (2022). Existing work that uses LLMs for generating action plans focuses on methods for grounding language in environment states Ichter et al. (2022); Huang et al. (2022b); Song et al. (2022b), or improving LLM plans through more structured output Singh et al. (2022); Liang et al. (2022b). In this work, we focus on using LLMs for exploration rather than directly generating action plans.

Tam et al. (2022) and Mu et al. (2022) recently demonstrated that language is a meaningful state abstraction when used for exploration. Additionally, Tam et al. (2022) experiment with using LLM latent representations of state descriptions for novelty exploration, relying on pretrained LLM encodings to detect novel textual states. To the best of our knowledge, we are the first to apply language-assisted decision-making to exploration by using LLMs to predict and verify environment dynamics through experience.

## 2.2 LANGUAGE GROUNDED IN INTERACTION

Without grounding, LLMs often fail to reason about real world dynamics Bisk et al. (2020). Instruction following tasks have been a popular testbed for language grounding Chevalier-Boisvert et al. (2019); Anderson et al. (2018); Ku et al. (2020); Shridhar et al. (2020) prompting many improvements to decision making conditioned on language instructions Yu et al. (2018); Lynch & Sermanet (2020); Nottingham et al. (2021); Suglia et al. (2021); Kuo et al. (2021); Zellers et al. (2021); Song et al. (2022a); Blukis et al. (2022). Other prior work used environment interactions to ground responses from question answering models in environment state Gordon et al. (2018); Das et al. (2018) or physics Liu et al. (2022). Finally, Ammanabrolu & Riedl (2021) learn a grounded textual world model from environment interactions to assist an RL agent in planning and action selection. In this work, our DECKARD agent also uses a type of textual world model but it is obtained few-shot from an LLM and then grounded in environment dynamics by verifying hypotheses through interaction.

## 2.3 MODULARITY IN RL

Modular RL proposes to learn several independent policies in a composable way to facilitate training and generalization Simpkins & Isbell (2019). Ammanabrolu et al. (2020) and Patil et al. (2020) demonstrate how modular policies can improve exploration by reducing policy horizons, the former using the text-based game Zork and the latter using Minecraft. We implement modularity for Minecraft by finetuning a pretrained transformer policy with adapters, a technique recently implemented for RL by Liang et al. (2022a) for multi-task robotic policies.

## 2.4 MINECRAFT

Minecraft is a vast open-ended world with complex dynamics and sparse rewards. Crafting items in the Minecraft technology tree has long been considered a challenging task for reinforcement learning, requiring agents to overcome extremely delayed rewards and difficult exploration Skrynnik et al. (2021); Patil et al. (2020); Hafner et al. (2023). This is partially due to the scarcity of items in the environment, but also due to the depth of some items in the game's technology tree. The purpose of our work is to overcome the latter of these two difficulties by better learning and navigating Minecraft's technology tree.

Several existing agents overcome the problem of item scarcity in Minecraft by simplifying environment parameters such as action duration Patil et al. (2020) or block break time Hafner et al. (2023), making comparison between methods difficult. For this reason we compare minimally to other Minecraft agents (see Table 2), focusing our evaluation on the benefits of LLM-guided exploration with DECKARD. We use the video pretrained (VPT) Minecraft agent Baker et al. (2022) as a starting

point for exploration and finetuning, and we use the Minedojo implementation of the Minecraft Environment Fan et al. (2022).

## 3 BACKGROUND

### 3.1 MODULAR REINFORCEMENT LEARNING

Rather than train a single policy with sparse rewards, modular RL advocates learning compositional policy modules Simpkins & Isbell (2019). DECKARD automatically discovers subgoals in Minecraft—each of which maps to an independently trained policy module—and learns a DAG of dependencies (the AWM) to transition between subgoals. Policy modules are trained in an environment modeled by a POMDP with states $s \in \mathcal{S}$, obseravtions $o \in O$, actions $a \in \mathcal{A}$, and environment dynamics $\mathcal{T} : \mathcal{S}, \mathcal{A} \rightarrow \mathcal{S}'$. These elements are common between modules, but each subgoal defines different initial states $\mathcal{S}_0$ and observations $O_0$, terminal states $\mathcal{S}_t$, and reward functions $\mathcal{R} : \mathcal{S}, \mathcal{A} \rightarrow \mathbb{R}$, according to the particular subgoal. $\mathcal{S}_0$ and $O_0$ are defined by the current subgoal's parents in the DAG, and $\mathcal{S}_t$ and $\mathcal{R}$ are defined by the current subgoal. For example, the *craft wooden pickaxe* subgoal has parents *craft planks* and *craft stick*, so $\mathcal{S}_0$ includes these items in the agent's starting inventory. This subgoal recieves a reward and terminates when a wooden pickaxe is added to the agent's inventory. Section 5 and Appendix B provide more details.

Due to the compositionality of modular RL, individual modules can be chained together to achieve complex tasks. In our case, given a goal state $s_g$, we use the subgoal DAG to create a path from our current state to $s_g$, $[s_0, s_1, ..., s_g]$, where each $s$ represents the terminal state for a subgoal. By chaining together sequences of subgoal modules, we can successfully navigate to connected portions of the currently discovered DAG and reach arbitrary goal states.

### 3.2 LARGE LANGUAGE MODELS

Large language models (LLM) are trained with a language modeling objective to maximize the likelihood of training data from large text corpora. As LLMs have grown in size and representational power, they have seen success on various downstream tasks by simply modifying their input, referred to as prompt engineering Brown et al. (2020). Recent applications of LLMs to decision-making have relied partially or entirely on prompt engineering for their action planning Ichter et al. (2022); Song et al. (2022b); Huang et al. (2022b); Singh et al. (2022); Liang et al. (2022b). We follow this pattern to extract knowledge from LLMs and construct our AWM. We prompt OpenAI's Codex model OpenAI (2022) to generate DECKARD's hypothesized AWM. Codex is trained to generate code samples from natural language. As with previous work Singh et al. (2022); Liang et al. (2022b), we find that structured code output works well for extracting knowledge from LLMs. We structure LLM output by prompting Codex for a python dictionary of Minecraft item dependencies, which we then map to a DAG with nodes and edges that represent items and dependencies between those items (see Section 5.1 and Appendix A).

## 4 DECKARD

### 4.1 ABSTRACT WORLD MODEL

Our method, **DEC**ision-making for **K**nowledgable **A**utonomous **R**einforcement-learning **D**reamers (DECKARD), builds an Abstract World Model (AWM) of subgoal dependencies from state abstractions. We begin by assuming a textual state representation function $\phi : O \rightarrow X$. Textual state representations $x \in X$ make up the nodes for our AWM $G : X, E$ with directed edges $E$ defining the dependencies between $X$. We further constrain $G$ to a directed acyclic graph (DAG) so that the nodes of the DAG represent subgoals useful in navigating towards a target goal. In our experiments, we use the agent's current inventory as $X$, a common component of the Minecraft observation space Fan et al. (2022); Hafner et al. (2023).

We update $G$ from agent experience through environment exploration. When the agent experiences node $x_t$ for the first time, $G$ is updated by adding edges between the previous node $x_{t-1}$ and the new node $x_t$. When trying to reach a previously experienced node, DECKARD recovers the path from

---

**Algorithm 1** DECKARD

---

$G \leftarrow LLM()$      // hypothesize AWM with LLM
$C \leftarrow X : 0$      // dict of visit counts
$V \leftarrow \emptyset$      // set of verified nodes
**while** $training$ **do**
   // Dream Phase
   $F \leftarrow Frontier(G, V)$
   **if** $any(C(F) \leq c_0)$ **then**
      $\bar{x} \leftarrow SampleBranch(F \mid C(F) \leq c_0)$
   **else**
      $\bar{x} \leftarrow SampleBranch(F \cup V)$
   **end if**
   // Wake Phase
   $x \leftarrow x_0$
   **for** $t = 1...|\bar{x}|$ **do**
      $x' \leftarrow ExecuteSubgoal(\bar{x}_t)$
      $C(x') \leftarrow C(x') + 1$
      **if** $x' \notin V$ **then**
         $G \leftarrow AddEdge(G, x, x')$
         $V \leftarrow V \cup \{x'\}$
      **end if**
      $x \leftarrow x'$
   **end for**
**end while**

---

current node $x_0$ to the target node $x_t$ from the AWM. DECKARD then executes policies for each recovered node until it reaches the target goal.

## 4.2 LLM Guidance

The setup so far (referred to in our experiments as "DECKARD (No LLM)") allows the construction of a modular RL policy for navigating subgoals. However, the agent is still learning the AWM tabula-rasa. The key insight of DECKARD is that we can hypothesize the AWM with knowledge from an LLM. We use in-context learning, as described in Section 5.1, to predict $G$ from an LLM with predicted edges, $\hat{E}$. While acting in the environment, we verify or correct edges of $G$ and track the set of nodes that have been verified $V$ thus grounding the AWM hypothesized by the LLM in environment dynamics.

### 4.2.1 Dream Phase

Equipped with a hypothesized AWM, we iterate between *Dream* and *Wake* phases for guided exploration toward a goal (see Algorithm 1). During the *Dream* phase, we compute the verified frontier $F$ of $G$, composed of verified nodes $V$, with predicted edges to unverified nodes $G - V$. In addition, if a path between $V$ and the current task's goal exists, $F$ is pruned to only include nodes along the predicted path to the goal. For example, after learning to craft *planks*, subgoals *door* and *stick* are potential frontier nodes. However, if the target item is *wooden pickaxe*, DECKARD will eliminate *door* as a candidate node for exploration since *stick* is part of the LLM-predicted recipe for the target item and *door* is not. Finally, we sample a branch $\bar{x}$ terminating with an element from $F$ to explore during the *Wake* phase. If all nodes in $F$ have been sampled at least $c_0$ times (where $c_0$ is an exploration hyperparameter) without success, we the sample from all $V$ rather than $F$ only.

### 4.2.2 Wake Phase

Next, during the *Wake* phase, the agent executes the sequence of subgoals $\bar{x}$ updating $G$ with learned experience and adding verified nodes to $V$. If sampled from $F$, the final node in $\bar{x}$ will be unlearned, allowing the agent to explore in an attempt to reach the unverified node. If successful, the AWM is updated and the new node is also added to $V$. When adding a newly verified node $x$ we begin

finetuning a new subgoal policy for $x$ (see Section 5). Beyond reducing the number of iterations it takes to construct $G$, one benefit of initializing $G$ with an LLM is that we do not finetune subgoals for nodes outside of the predicted path to our target goal. If the predicted recipes fail, then DECKARD begins training additional subgoal policies to assist in exploration. This drastically reduces the number of environment steps required to train DECKARD.

## 5 EXPERIMENT SETUP

We apply DECKARD to crafting items in Minecraft, an embodied learning environment that requires agents to perform sequences of subgoals with sparse rewards. Our agent maps inventory items to AWM subgoals and learns a modular policy that can be composed to achieve complex tasks. By learning modular policies, our agent is able to collect and craft arbitrary items in the Minecraft technology tree.

### 5.1 PREDICTING THE ABSTRACT WORLD MODEL

In our experiments, we predict the AWM using OpenAI's Codex model OpenAI (2022) by prompting the LLM to generate recipes for Minecraft items. We prompt Codex to "Create a nested python dictionary containing crafting recipes and requirements for minecraft items" along with additional instructions about the dictionary contents and two examples: *diamond pickaxe* and *diamond* (see Appendix A). We iterate over 391 Minecraft items, generating recipes as well as tool requirements (mining stone requires a pickaxe) and workbench requirements (crafting a pickaxe requires a crafting table). The hypothesized AWM is generated at the start of training, so no forward passes of the LLM are necessary during training or inference. Table 1 shows the accuracy of the hypothesized un-grounded AWM.

| Metric | All Items | Tools Only |
|---|---|---|
| Collectable vs. Craftable | 57 | 100 |
| Crafting Table / Furnace | 84 | 96 |
| Recipe Correct Items | 66 | 81 |
| Recipe Exact Match | 55 | 69 |

Table 1: LLM accuracy when predicting various node features: whether an item is collectable (no parents) or craftable (has a recipe), whether it requires a crafting table or furnace to craft, whether recipe ingredients are correct, and whether the recipe is an exact match (including ingredient quantities). The first results column includes all 391 Minecraft items, whereas the second column only includes the 37 items in the tool technology tree.

### 5.2 SUBGOAL FINETUNING

Rather than train each module from scratch, we finetune transformer adapters for each module with an RL objective following the adapter architecture from Houlsby et al. (2019). We use the Video-Pretrained (VPT) Minecraft model as our starting policy Baker et al. (2022). We chose to finetune VPT as it proved to be more sample efficient and more stable than training policies from scratch. Moreover, since VPT is pretrained on a variety of Minecraft skills, the non-finetuned VPT model explores the environment more thoroughly than a random agent. Our implementation of VPT finetuned with adapters is referred to as VPT-a.

Adapters are especially well suited for modular finetuning due to their lightweight architecture Liang et al. (2022a). In our agent, each subgoal module corresponds to one set of adapters and only contains 9.5 million trainable parameters, approximately 2% of the 0.5 billion parameter VPT model. This allows us to train a separate set of adapters for each subgoal and still keep all parameters in memory concurrently, a practical benefit of using adapters for modular, compositional RL policies.

### 5.3 ENVIRONMENT DETAILS

We use Minedojo's Minecraft implementation for our experiments Fan et al. (2022). As with VPT Baker et al. (2022), our subgoal policies use a pixel only observation space and a large multi-discrete

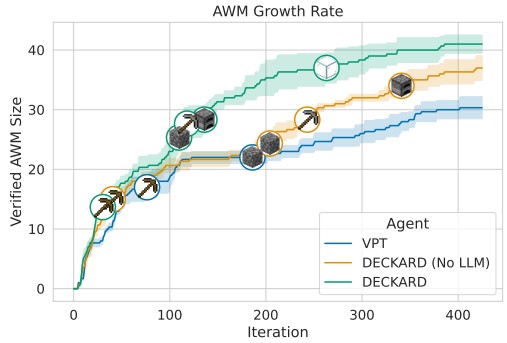
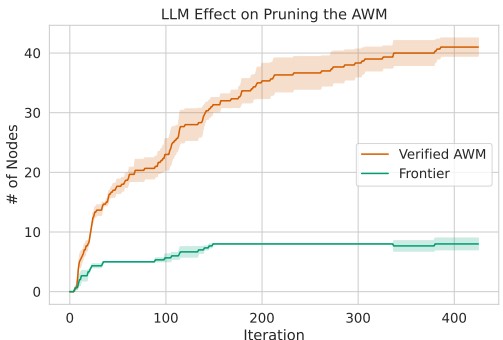

(a) Rate of exploration for during open-ended exploration, measured by the size of the verified AWM per iteration. Each iteration includes one *Dream* and one *Wake* phase. *VPT* measures the number of items discovered by a non-finetuned VPT policy and *No LLM* ablates LLM guidance. LLM guidance more than halves the time it takes to discover difficult items such as stone tools and glass.

(b) DECKARD prunes the AWM by only sampling from the frontier of verified and hypothesized AWM nodes. Without LLM guidance, our agent would sample from the entire AWM during exploration. However, the AWM grows in size throughout training and many nodes become dead ends, slowing exploration.

Figure 2: Open-ended exploration results.

action space, while our overall policy transitions between subgoals based on the agent's current inventory. Unlike VPT, we use standard high-level crafting actions that instantly crafts target items from inventory items. At the time of this writing, Minedojo does not support the VPT style of human-like crafting in a GUI, so we instead remove the VPT action for opening the crafting GUI and replace it with 254 crafting actions (one for each item). This brings our multi-discrete action space to 121 camera actions and 8714 keyboard/crafting actions, and our observation space to 128x128x3 pixels plus 391 inventory item quantities (only used to transition between subgoals).

Because our subgoals map to individual items, there is an intrinsic separation between items that are collected from the environment versus those that require crafting. While we must finetune a set of adapters for subgoals that require navigating or collecting items from the environment, crafting subgoal policies map to a single craft action—making them much more space and sample efficient compared to collectable item subgoals.

## 5.4 EXPERIMENTS

We evaluate DECKARD on both crafting tasks—in which the agent learns to collect ingredients and craft a target item—and open-ended exploration. In open-ended exploration, although there is no extrinsic learning signal, DECKARD is intrinsically motivated to explore new AWM nodes. We compare the growth of the agent's verified AWM during open-ended exploration for DECKARD with and without LLM guidance along with a VPT baseline. Next, we compare LLM-guided DECKARD to RL baselines and DECKARD without LLM guidance on goal-driven tasks for collecting/crafting: logs, wooden pickaxes, cobblestone, stone pickaxes, furnaces, sand, and glass. We also compare to several popular Mincraft agents on the "craft a stone pickaxe task" (see Table 2). Finally, we evaluate the effect of artificial errors in the hypothesized AWM to simulate errors in LLM output and demonstrate DECKARD's robustness to LLM accuracy.

## 6 EXPERIMENT RESULTS

### 6.1 OPEN-ENDED EXPLORATION

DECKARDis intrinsically motivated to explore new nodes, always sampling and attempting to craft new items, and thus does not require a target task to improve exploration. We can measure the effect of DECKARD on exploration by tracking the growth of the agent's verified AWM nodes. Figure 2a shows the speed of exploration when using DECKARD with and without LLM guidance. We

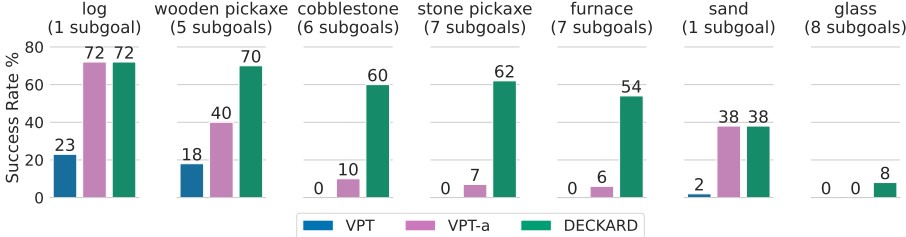

Figure 3: Success rates for item tasks on random world seeds. The VPT agent shows success rates of the pretrained VPT policy without any additional finetuning. VPT-a finetunes VPT using the same training setup as DECKARD without modularity or LLM guidance. As indicated by the results for *log* and *sand* (item tasks composed of a single subgoal), VPT-a is equivalent to a single subgoal policy. DECKARD without LLM-guidance has the same success rate as the full DECKARD agent.

also compare DECKARD to a VPT baseline that explores the environment without an AWM with a non-finetuned VPT policy. Although VPT does not construct an AWM, it gathers Minecraft items and randomly attempts to craft new items from the gathered ingredients. We track how many items it has discovered and plot that quantity in Figure 2a. DECKARD without LLM guidance constructs an AWM from scratch, but only the LLM-guided DECKARD agent uses LLM guidance to decide which items to collect and which recipes to attempt next. Note that DECKARD subgoal policies are initialized with VPT, so VPT starts out exploring at a similar rate to DECKARD.

The DECKARD and VPT agents quickly learn to mine logs and craft wooden items. However, one exploration hurdle is discovering that wooden pickaxes are a prerequisite for mining cobblestone. As seen in Figure 2a, it takes DECKARD without LLM guidance and the VPT baseline 2x and 3x longer respectively to learn to use a wooden pickaxe to mine cobblestone. Once the agents learn how to mine cobblestone, they can begin adding stone items to their AWM. However, only DECKARD avoids oversampling dead ends in the crafting tree allowing it to quickly explore new states. Also, the LLM incorrectly predicts that glass can be collected without crafting or tools of any kind, but DECKARD overcomes and corrects this error, successfully crafting glass and adding the correct recipe to the AWM.

In general, the frontier $F$ of the verified AWM nodes $V$ is much smaller than $G$. This difference increases as the agent continues to explore and add verified nodes to $G$. Figure 2b shows the sizes of $G$ and $F$ throughout open-ended exploration for DECKARD. The smaller size of $F$ means that each iteration DECKARD is more likely to sample items that are useful for crafting something new. Eventually, difficult to reach or erroneous nodes in $F$ could limit exploration. This is the reason we stop prioritizing sampling from the frontier after $c_0$ failed attempts to reach nodes from $F$. However, through continued exploration, DECKARD can discover and correct erronously predicted nodes.

## 6.2 CRAFTING TASKS

We also evaluate DECKARD on tasks that require collecting or crafting a specific item. Rather than sample from the entire frontier $F$ as with open-ended exploration, we only sample nodes from $F$ predicted to lead to the target item.

Figure 3 compares DECKARD success rates to baselines across item tasks: logs, wooden and stone pickaxes, cobblestone, furnace, sand, and glass. The VPT baseline is the non-finetuned VPT policy acting in the environment, and VPT-a follows the same training setup as our subgoal policies (see Section 5.2). Agents are allowed a maximum of 1,000 environment steps to obtain collectable items (log and sand), and 5,000 steps for all other craftable items. Training for each agent is limited to 6 million steps, although DECKARD only takes that many for the "craft glass" task. DECKARD outperforms directly training on item tasks with a traditional reinforcement learning signal and learns to craft items further up the technology tree where the baseline completely fails.

Note that we use random world seeds for all evaluation making scarce items more difficult to reliably collect. For example, the fact that sand is more rare than logs is reflected in their respective success

| Method | Demos | Dense Rewards | Observations | Actions | Steps |
|---|---|---|---|---|---|
| Align-RUDDER Patil et al. (2020) | Expert | ✗ | Pixels & Meta | 61 | 2M |
| VPT+RL Baker et al. (2022) | Videos | ✓ | Pixels Only | 121, 8461 | 2.4B |
| DreamerV3 Hafner et al. (2023) | None | ✓ | Pixels & Meta | 25 | 6M |
| DECKARD (No LLM) | Videos | ✗ | Pixels & Inventory | 121, 8714 | 32M |
| DECKARD | Videos | ✗ | Pixels & Inventory | 121, 8714 | 2.6M |

Table 2: Direct comparison between minecraft agents is difficult because of the various shortcuts used to solve the difficult exploration task. Align-RUDDER, relies on expert demonstrations. DreamerV3 and Align-RUDDER, simplify the vast combinatorial action space. VPT+RL and DreamerV3 provide intermediate rewards that require knowledge of Minecraft's crafting tree. The final column above compares how long each method takes to learn the "craft stone pickaxe" task. Despite its challenging learning setup, DECKARD achieves sample efficiency equal to or better than existing agents.

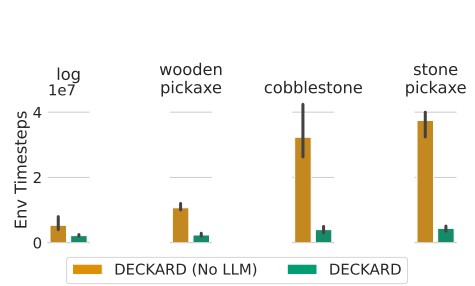

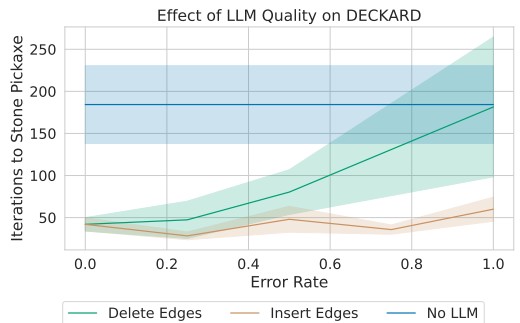

(a) Environment steps until the discovery of target items. LLM guidance improves sample efficiency by an order of magnitude by only learning subgoal policies for the path to the target item predicted by the LLM.

(b) Effect of errors in the LLM predicted AWM, measured by the number of iterations until DECKARD learns to craft a stone pickaxe. Starting from a ground truth graph, node edges are deleted/inserted to simulate LLM quality.

Figure 4

rates in Figure 3. Also, items that depend on logs (pickaxes, cobblestone, furnace) and sand (glass) will have success rates bounded by that of their parent nodes in the technology tree.

The sample efficiency of DECKARD is especially notable when applied to task-conditioned LLM guidance. With LLM guidance, DECKARD can avoid learning subgoal policies for items it predicts are unnecessary for the current goal (see Section 4.2.2). Figure 4a demonstrates the difference that only training policies for predicted subgoals can make on sample efficiency. Without LLM guidance, DECKARD finetunes subgoal policies for an average of fifteen different collectable items. With guidance, DECKARD only finetunes subgoal policies for collecting needed items (such as logs and cobblestone when crafting a stone pickaxe)—resulting in an order of magnitude improvement in sample efficiency.

Although not the primary goal of this work, we compare DECKARD to several agents trained to craft items along the Minecraft technology tree. Table 2 includes a high level overview of these agents and provides the number of environment samples for each to learn the "craft stone pickaxe" task. Note that each of these agents uses vastly different action and observation spaces as well as pretraining data. For example, DECKARD does not require any reward shaping from domain expertise, expert demonstrations, or simplifications of the observation and action spaces. Despite this, Table 2 shows that DECKARD's sample efficiency is equal to or better than that of previous work, improving by an order of magnitude or more for agents with comparable action spaces.

## 6.3 ROBUSTNESS

Finally, we evaluate our claim that DECKARD is robust to errors in LLM output. While LLMs are becoming surprisingly accurate when answering specific questions about niche domains such as

Minecraft, they are not grounded in environment knowledge and sometimes output erroneous facts Valmeekam et al. (2022). Figure 4b shows training time for DECKARD on the target task "craft stone pickaxe" for various error types and rates in the hypothesized AWM. For each run, we start with a ground truth AWM and programatically introduce the indicated errors over at least three different random seeds for each error type and rate.

While the most common error that our LLM-predicted AWM had was in the quantity of each ingredient (see Table 1), we found that DECKARD was robust to this error and often ended up with a surplus of ingredients regardless. Figure 4b shows the effect of inserting and deleting edges in the AWM. Inserted edges always add sand as an ingredient for the current item, and deleted edges may remove recipe ingredients or a required tool/crafting table. The wide error bands in Figure 4b indicate that certain edges in the AWM have a bigger influence on exploration when inserted/deleted. Despite this, DECKARD with LLM guidance successfully outperforms DECKARD without LLM guidance even when faced with large errors in LLM output, demonstrating DECKARD's robustness to LLM output as an exploration method.

## 7 DISCUSSION & CONCLUSION

In line with proposals to utilize pretrained models in RL Agarwal et al. (2022), we extract knowledge from LLMs in the form of an Abstract World Model (AWM) that defines transitions between subgoals in a directed acyclic graph. Our agent, DECKARD (**DEC**ision-making for **K**nowledgable **A**utonomous **R**einforcement-learning **D**reamers), successfully uses the AWM to intelligently explore Minecraft, learning to craft arbitrary items through a modular RL policy. Initializing DECKARD with an LLM-predicted AWM improves sample efficiency by an order of magnitude. Additionally, we use environment dynamics to ground the hypothesized AWM by verifying and correcting it with agent experience, robustly applying large-scale, noisy knowledge sources to aid in sequential decision-making.

We, along with many others, hope to utilize the potential of LLMs for unlocking internet-scale knowledge for decision-making. Throughout this effort, we encourage the pursuit of robust and generalizable methods, like DECKARD. One drawback of DECKARD, along with many other LLM-assisted RL methods, is that it requires an environment already be grounded in language. Some preliminary methods for generating state descriptions from images are used by Tam et al. (2022), but this remains an open area of research. Additionally, we assume an abstraction over environment states to make predicting dependencies scalable. We leave the problem of of automatically identifying state abstractions to future work. Finally, DECKARD considers only deterministic transitions in the AWM. While a similar approach to ours could be applied to stochastic AWMs, that is out of the scope of this work.

DECKARD introduces a powerful approach for exploration using LLMs for guidance. By alternating between sampling predicted states on the frontier of what has been discovered (The *Dream* phase) and executing subgoals to arrive there (The *Wake* phase), we successfully apply noisy LLM world knowledge to an Abstact World Model over subgoals.

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

## A  CODEX IN-CONTEXT LEARNING

### A.1  PROMPTING DETAILS

We use OpenAI's Codex model (code-davinci-002) OpenAI (2022) to predict an Abstract World Model (AWM) for DECKARD. We prompt the model with instructions in code comments that instruct the model to generate a python dictionary with information for Minecraft item requirements. We also provide example entries for "diamond pickaxe" and "diamond". We then iterate over all 391 Minecraft items to generate the next entry in the python dictionary. We organize the data in the dictionary entries into the following item attributes:

- requires_crafting_table: whether an item requires the agent to have a crafting table prior to crafting
- requires_furnace: whether the item is smelted with a furnace
- required_tool: what tool is required to collect the item from the environment
- recipe: list of ingredients and ingredient quantities to craft the item

The full prompt we use can be found below:

```
# Create a nested python dictionary containing crafting recipes and
requirements for minecraft items.
# Each crafting item should have a recipe and booleans indicating
whether a furnace or crafting table is required.
# Non craftable blocks should have their recipe set to an empty list
and indicate which tool is required to mine.

minecraft_info = {
    "diamond_pickaxe": {
        "requires_crafting_table": True,
        "requires_furnace": False,
        "required_tool": None,
        "recipe": [
            {
                "item": "stick",
                "quantity": "2"
            },
            {
                "item": "diamond",
                "quantity": "3"
            }
        ]
    },
    "diamond": {
        "requires_crafting_table": False,
        "requires_furnace": False,
        "required_tool": "iron_pickaxe",
        "recipe": []
    },
    "[insert item name]": {
```

### A.2  PARSING DETAILS

When parsing output, we consider any item with a recipe of length zero to be a collectable item (it will have no parents in the AWM). In our experiments, Codex generated parsable entries for all but one Minecraft item (brown mushroom block).

In general, Codex predicts the same item identifier that Minedojo Fan et al. (2022) uses. One major exception is that of *planks*, a common item essential for many recipes. We parse *plank* and *wood* as

well as any variant of these two (*oak plank*) as *planks*. We also parse *cane* as *reeds*. Note that in all these cases the predicted names are also common identifiers for these items in minecraft, but they do not match the Minedojo identifiers.

Finally, we remove circular dependencies from the predicted AWM. First we remove edges from crafting table, furnace, and tool nodes to items that are found in the recipes for those nodes. Then we remove edges both to and from items found in eachother's recipes. There were four cases of circular dependencies in our hypothesized AWM, between planks and crafting table, log and wooden axe, fermented spider eye and spider eye, and purpur block and purpur pillar.

## A.3 ADDITIONAL RESULTS

| "Tool Only" Items | | | |
|---|---|---|---|
| coal | furnace | crafting_table | log |
| planks | stick | cobblestone | iron_ore |
| iron_ingot | gold_ore | gold_ingot | diamond |
| wooden_hoe | wooden_sword | wooden_axe | wooden_pickaxe |
| wooden_shovel | stone_hoe | stone_sword | stone_axe |
| stone_pickaxe | stone_shovel | iron_hoe | iron_sword |
| iron_axe | iron_pickaxe | iron_shovel | golden_hoe |
| golden_sword | golden_axe | golden_pickaxe | golden_shovel |
| diamond_hoe | diamond_sword | diamond_axe | diamond_pickaxe |
| diamond_shovel | | | |

Table 3: The 37 Minecraft items from the tool technology tree.

| Metric | All Items | Tools Only |
|---|---|---|
| Accuracy: Collectable vs. Craftable Label | 57 | 100 |
| Accuracy: Workbench (Crafting Table/Furnace) | 84 | 96 |
| Accuracy: Recipe Ingredients | 66 | 81 |
| Accuracy: Recipe Ingredients & Quantities | 55 | 69 |
| % Items w/ Incorrectly Inserted Dependencies | 42 | 8 |
| % Items w/ Missing Dependencies | 35 | 26 |
| Standard Deviation In Predicted Ingredient Quantity | 0.98 | 0.34 |
| Absolute Error In Predicted Ingredient Quantity | 2.77 | 1.50 |
| Average Error In Predicted Ingredient Quantity | -1.07 | 0.50 |

Table 4: Additional Codex metrics for predicting the Minecraft AWM.

Our experiments with few-shot prompting Codex to generate the AWM for Minecraft show that LLMs can generate structured knowledge for decision making. However, predictions are not perfect, so we treat them as hypotheses that are verified by environment interactions. Codex does perform better on the tool technology tree, items that are both more common and more relevant for crafting agents. A large percentage of errors also appears to be from incorrectly predicted ingredient quantities.

# B SUBGOAL FINETUNING

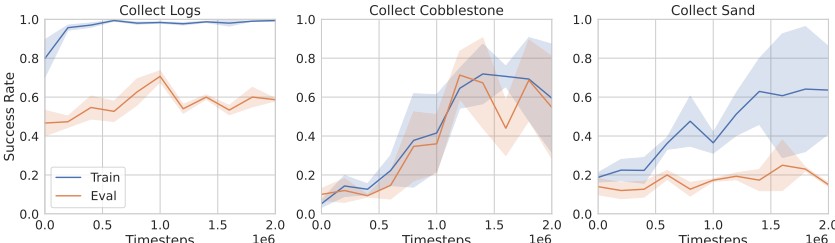

Figure 5: Results for finetuning subgoal policies. DECKARD's VPT-based subgoal policies are trained on seeds where the target item is nearby and evaluated on random world seeds. Of these results, cobblestone is the most ubiquitous and sand the least, as indicated by how well the policy generalizes to random Minecraft world seeds.

| Hyperparameter | Value |
|---|---|
| VPT Checkpoint | bc-house-3x |
| Environment Steps per Actor per Iteration | 500 |
| Number of Actors | 4 |
| Batch Size | 40 |
| Iteration Epochs | 5 |
| Learning Rate | 0.0001 |
| $\gamma$ | 0.999 |
| Value Loss Coefficient | 1 |
| Initial KL Loss Coefficient | .1 |
| KL Coefficient Decay per Iteration | .999 |
| Adapter Downsize Factor | 16 |

Table 5: DECKARD subgoal finetuning hyperparameters.

## B.1 VPT FINETUNING

We finetune VPT (3x w/ behavior cloning on house contractor data) Baker et al. (2022) with reinforcement learning (RL) using transformer adapters as described by Houlsby et al. (2019). That is, we insert two adapter modules with residual connections in each transformer layer, with a 16x reduction in hidden state size. We updated the adapters and agent value head using proximal policy optimization (PPO) Schulman et al. (2017), but we leave the rest of the agent unchanged (including the policy head).

Following Baker et al. (2022), we replace the traditional entropy loss in the PPO algorithm with a KL loss between the current policy and the non-finetuned VPT policy. The purpose of this loss is to prevent catastrophic forgetting early in training. Our experiments reaffirmed the importance of this term, even when leaving the majority of the VPT weights unchanged. The KL loss coefficient decays throughout training to allow the agent to reach an optimal policy.

## B.2 MINECLIP REWARD

Along with their Minedojo Minecraft implementation, Fan et al. (2022) introduced a text and video alignment model for Minecraft called MineClip and showed how the model could be used for automatic reward shaping given a text goal. We use MineClip to provide reward shaping for finetuning DECKARD subgoal and VPT-a policies. Unlike Fan et al. (2022), we implement MineClip reward shaping by subtracting $clip_{low} = 21$ from the MineClip alignment score and scaling by $clip_\alpha = 0.005$, smoothed over $smooth = 50$ steps:

$$reward_{clip} = clip_\alpha \times max(0, mean(score\_buffer_{-smooth:}) - clip_{low})$$

Additionally, we only provide the agent with non-zero reward when the MineClip alignment score reaches a new maximum for the episode. Finally we provide a reward of +1 when the agent successfully adds the target item to its inventory.

### B.3 MINECRAFT SETTINGS

Use use the Minedojo simulator Fan et al. (2022) with the "creative" metatask for our experiments. We found Minedojo preferable to MineRL Guss et al. (2019), due to a reduced tendency to crash when running many parallel environment instances. We followed the VPT Baker et al. (2022) observation and action spaces—128x128x3 pixel observation space and 121x8461 multi-discrete action space—with the modification of replacing the "open inventory" action with 254 discrete crafting actions.

When training subgoal policies, we initialize the agent with items from the current node's parents. For example, when training the *collect cobblestone* subgoal, we initialize the agent with a wooden pickaxe, the required tool for cobblestone in the AWM. We terminate each episode after 1,000 environment steps, generating a new world.

We also found that finetuning was sensitive to world seed when training. For example, many world seeds spawned the agent far from target items, stranded on islands, or underwater. To mitigate the effect of poor world initialization on training, we use a single world seed for training each subgoal policy and then evaluate on random world seeds. We find that VPT is able to generalize to random seeds after training on a training seed.

## C ABSTRACT WORLD MODEL

In many environments, multiple possible transitions between subgoals may exist. For example, in Minecraft, an agent can obtain coal through mining or by burning wood in a furnace. Ideally, edges of the AWM would provide paths with high success rate to each node. In our implementation we keep the first experienced edge between nodes, assuming it to be the simplest path.

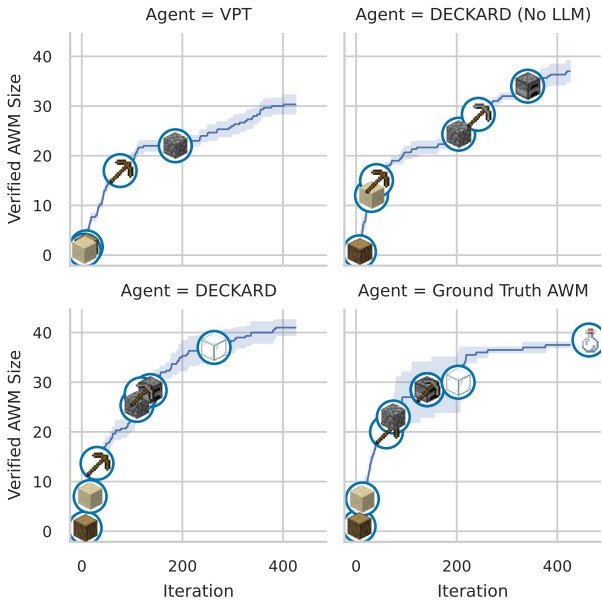

Figure 6: AWM growth during open-ended exploration. The first three quadrants are identical to Figure 2a. The last quadrant adds results for a ground truth AWM. The agent learns to craft glass much sooner and also learns to craft glass bottles, and item none of the other methods reached.

