# OpenReview forum: "Do Embodied Agents Dream of Pixelated Sheep?: Embodied Decision Making using Language Guided World Modelling"
_ICLR.cc/2023/Workshop/RRL — RRL 2023 Oral_

### Official Review · Reviewer_QeW8 · 2023-02-25

**Rating:** 3
**Confidence:** 4

**Review:**

The authors propose a method that uses a large language model in a few-shot manner to improve exploration in a Minecraft environment. The proposed method is divided in two parts: (i) a "Dream" phase in which planning and execution is done using the Abstract World Model (AWM); (ii) a "Wake" phase in which the agent learns a modular policy for each subgoal. The results show that the proposed model improve exploration and enhance performance over a Video-Pretrained (VPT) Minecraft policy (the starting policy).

Strengths:

1. The modular policy proposed by the authors is interesting and the same idea can be applied to other environments.
2. The paper is a very good fit for the workshop. It uses a VPT as the starting policy, and it uses an LLM in a few-shot manner as a world model to improve exploration.
3. The results show that the proposed method improve exploration and sampling efficiency in a challenging environment: Minecraft.

Weaknesses:

1. The Abstract World Model created based on the LLM model still seems really specific (what is reasonable and expected) but hard to generalize the idea for other challenging environments. The authors add a small discussion in Section 7, but a broad discussion on fine-tuning or training target-specific fine-tuned language models to be combined with the proposed method for other environments would be helpful for readers.
2. (Minor): Correct typos in the manuscript. Ex: “Abstract” in the last line of Section 7.

Overall, the reviewer thinks that the work is a good fit for the workshop and can bring fruitful discussion regarding: (i) improving pre-trained policies; (ii) using pre-trained LLMs as world models to improve exploration.

---

### Official Review · Reviewer_bDWa · 2023-02-27
**Good paper that reuses pretrained LLM to guide agent exploration**

**Rating:** 4
**Confidence:** 3

**Review:**

The authors propose using pre-trained large language models (LLMs) to hypothesize an abstract world model for embodied RL agents. The RL agent uses this abstract world model to guide its exploration. The method, DECKARD, operates in two phases: 1) the Dream phase, where the LLM suggests subgoals, and 2) the Wake phase, where the agent verifies or corrects its model based on its experience. Unlike other LLM for RL papers, this work uses LLMs only to propose a world model but uses its own grounded experience to learn from. The authors present this work in the context of item crafting in Minecraft and they use OpenAI Codex as the pre-trained LLM.

The writing is very clear and the limitations of the method are discussed. The method is simple and effective. Large language models take tremendous computing and data resources to train; the potentially valuable knowledge in them about what information to inquire about and how to complete tasks can benefit RL agents. This makes this paper a relevant submission for this workshop.